# EMERGENT COMMUNICATION WITH ATTENTION

## ABSTRACT

To develop computational agents that better communicates using their own emergent language, we endow the agents with an ability to focus their attention on particular concepts in the environment. Humans often understand a thing or scene as a composite of concepts and those concepts are further mapped onto words. We implement this intuition as attention mechanisms in Speaker and Listener agents in a referential game and show attention leads to more compositional and interpretable emergent language. We also demonstrate how attention helps us understand the learned communication protocol by investigating the attention weights associated with each message symbol and the alignment of attention weights between Speaker and Listener agents. Overall, our results suggest that attention is a promising mechanism for developing more human-like emergent language.

## 1 INTRODUCTION

We endow computational agents with an ability to focus their attention on a particular concept in the environment and communicate using emergent language. Emergent language refers to the communication protocol based on discrete symbols developed by agents to solve a specific task (Nowak & Krakauer, 1999; Lazaridou et al., 2017). One important goal in the study of emergent language is clarifying conditions that lead to more compositional languages. Seeking compositionality provides insights into the origin of the compositional natures of human language and helps develop efficient communication protocols in multi-agent systems that are interpretable by humans.

Much recent work studies emergent language with generic deep agents (Lazaridou & Baroni, 2020) with minimum assumptions on the inductive bias of the model architecture. For example, a typical speaker agent encodes information into a single fixed-length vector to initialize the hidden state of a RNN decoder and generate symbols (Lazaridou et al., 2017; Mordatch & Abbeel, 2018; Ren et al., 2020). Only a few studies have explored some architectural variations (Słowik et al., 2020) to improve the compositionality of emergent language, and there remains much to be discussed about the effects of the inductive bias provided by different architectures.

We posit that towards more human-like emergent language we need to explore other modeling choices that reflect the human cognitive process. In this study, we focus on the attention mechanism. Attention is one of the most successful neural network architectures (Bahdanau et al., 2015; Xu et al., 2015; Vaswani et al., 2017) that have an analogy in psychology (Lindsay, 2020). The conceptual core of attention is an adaptive control of limited resources, and we hypothesize that this creates pressure for learning more compositional emergent languages. Compositionality entails a whole consisting of subparts. Attention allows the agents to dynamically highlight different subparts of an object when producing/understanding each symbol, which potentially results in clear associations between the object attributes and symbols.

Another reason to explore the attention mechanism is its interpretability. Emergent language is optimized for task success and the learned communication protocol often results in counter-intuitive and opaque encoding (Bouchacourt & Baroni, 2018). Several metrics have been proposed to measure specific characteristics of emergent language (Brighton & Kirby, 2006; Lowe et al., 2019) but these metrics provide rather a holistic view of emergent language and do not tell us a fine-grained view of what each symbol is meant for or understood as. Attention weights, on the other hand, have been shown to provide insights into the basis of the network's prediction (Bahdanau et al., 2015; Xu et al., 2015; Yang et al., 2016). Incorporating attention in the process of symbol production/comprehension will allow us to inspect the meaning of each symbol in the messages.

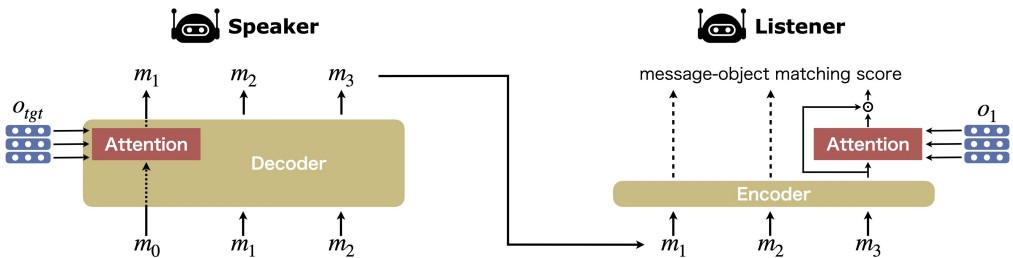

Figure 1: Illustration of the attention agents in the referential game.

In this paper, we test attention agents with the referential game (Lewis, 1969; Lazaridou et al., 2017), which involves two agents: *Speaker* and *Listener*. The goal of the game is to convey the type of object that Speaker sees to Listener. We conduct extensive experiments with two types of agent architectures, LSTM and Transformer, and two types of environments: the one-hot game and the Fashion-MNIST game. We compare the attention agents against their non-attention counterparts to show that adding attention mechanisms with disentangled inputs to either/both Speaker or/and Listener helps develop a more compositional language. We also analyze the attention weights and explore how they can help us understand the learned language. We visualize the learned symbol-concept mapping and demonstrate how the emergent language can deviate from human language and, with experiments with pixel images, how the emergent language can be affected by the visual similarity of referents. We also investigate the potential of the alignment between Speaker's and Listener's attention weights as a proxy for the establishment of common understanding and show that the alignment correlates with task success.

## 2   EXPERIMENTAL FRAMEWORK

### 2.1   REFERENTIAL GAME

We study emergent language in the referential game (Lewis, 1969; Lazaridou et al., 2017). The game focuses on the most basic feature of language, referring to things. The version of the referential game we adopt in this paper is structured as follows:

1. Speaker is presented with a target object $o_{tgt} \in \mathcal{O}$ and generates a message $m$ that consists of a sequence of discrete symbols.
2. Listener receives the message $m$ and a candidate set $C = \{o_1, o_2, ..., o_{|C|}\}$ including the target object $o_{tgt}$ and distractor objects sampled randomly without replacement from $\mathcal{O}$.
3. Listener chooses one object from the candidate set and if it is the target object, the game is considered successful.

The objects can be represented as a set of attributes. The focus of this game is whether agents can represent the objects in a compositional message that is supposedly based on the attributes.

### 2.2   AGENT ARCHITECTURES

Our goal in this paper is to test the effect of the attention mechanism on emergent language. The attention mechanism in its general form takes a query vector $\mathbf{x}$ and key-value vectors $\{\mathbf{y}_1, ..., \mathbf{y}_L\}$. The key-value vectors are optionally transformed into key vectors and value vectors or used as is. Then, attention scores $\{s_1, ..., s_L\}$ are calculated as the similarity between the query vector and the key vectors to the attention weights via the softmax function. Finally, the attention weights are used to produce a weighted sum of the value vectors. A key feature of attention is that it allows the agents to selectively attend to a part of disentangled vector inputs. Our intuition is that modeling direct associations between the symbol representations as query and disentangled input representations as key-value will bias agents toward packing each symbol with the information of a meaningful subpart of the inputs rather than with opaque and non-compositional information. To test these hypotheses, we design non-attention and attention agents for both Speaker and Listener (Figure 1).

### 2.2.1 OBJECT ENCODER

The objects are presented to the agents in the form of a set of real-valued vectors, e.g., attribute-wise one-hot vectors or patch-wise pretrained CNN feature vectors, as $\{\mathbf{o}^1, ..., \mathbf{o}^A\}$.

All the Speaker and Listener agents have their individual object encoders. The input vectors are independently linear-transformed into the size of the agent's hidden size and then successively go through the `gelu` activation (Hendrycks & Gimpel, 2016). The baseline non-attention agents average the transformed vectors into a single vector $\hat{\mathbf{o}}$ for the subsequent computations, whereas the attention agents leave the transformed vectors intact and attend to the set of vectors $\{\hat{\mathbf{o}}^1, ..., \hat{\mathbf{o}}^A\}$.

### 2.2.2 SPEAKER AGENTS

Speaker has an auto-regressive message decoder that takes the encoded vector(s) of the target object as input and generates a multi-symbol message $m = (m_1, ..., m_T)$. There is a number of possible choices for the decoder architecture with attention. To provide extensive empirical evidence on the effect of attention mechanism, we experiment with two common decoder architectures: the **LSTM** decoder from Luong et al. (2015) and the **Transformer** decoder from Vaswani et al. (2017).

At each time step $t$, the decoders embed a previously generated symbol into a vector $\mathbf{m}_{t-1}$ and produce an output hidden vector through three steps: (1) recurrent computation; (2) attention; (3) post-processing. As the decoders basically follow the original architecture, we only briefly describe each step in the LSTM and Transformer decoder with emphasis on how attention is incorporated. The more detailed descriptions can be found in Appendix C). The recurrent computation updates the input vector with the contextual information of previous inputs. The LSTM decoder uses a LSTM cell (Hochreiter & Schmidhuber, 1997) and the Transformer decoder uses the self-attention mechanism (Vaswani et al., 2017).

Then, with the contextualized input vector as the query $\mathbf{x}_t$ and the object vectors as the key-value vectors $\{\hat{\mathbf{o}}^1, ..., \hat{\mathbf{o}}^A\}$, the decoders perform attention. The LSTM decoder uses the bilinear attention, where the attention score $s_i$ is computed as $s_t^i = \mathbf{x}_t^\top \mathbf{W}_b \hat{\mathbf{o}}^i$, where $\mathbf{W}_b$ is a learnable matrix. The attention vector is calculated as the weighted sum of the original key-value vectors. The Transformer attention fist linear-transforms the input vector as $\mathbf{q}_t = \mathbf{W}_q \mathbf{x}_t, \mathbf{k}^i = \mathbf{W}_k \hat{\mathbf{o}}^i, \mathbf{v}^i = \mathbf{W}_v \hat{\mathbf{o}}^i$, where $\mathbf{W}_q, \mathbf{W}_k, \mathbf{W}_v$ are learnable matrices. Then the attention score is calculated using the scaled dot attention: $s_t^i = (\mathbf{q}_t^\top \mathbf{k}^i)/\sqrt{d}$, where $d$ is the dimension of the query and key vectors. Finally, the attention vector is calculated as the weighted sum of the value vectors $\mathbf{v}^i$. The original Transformer also has the multi-head attention mechanism, but in the main experiments, we set the number of the attention heads to one for interpretability and ease of analysis. The effect of multi-head attention is investigated in Appendix E.

Finally, in the post-processing step, the original query vector $\mathbf{x}_t$ and attended vector $\hat{\mathbf{x}}_t$ are integrated to produce the hidden vector to predict the next symbol.

**Non-attention (NoAT) Speaker** is a baseline agent that encodes the target object into a single vector $\hat{\mathbf{o}}_{tgt}$. The source-target attention in the decoder always attends to that single vector and does not change where to focus during message generation.

**Attention (AT) Speaker**, in contrast, encodes the target object into a set of vectors $\{\hat{\mathbf{o}}_{tgt}^1, ..., \hat{\mathbf{o}}_{tgt}^A\}$ and the source-target attention dynamically changes its focus at each time step.

Our agent design is motivated by a fair comparison of non-attention and attention agents. They have the same architecture and number of parameters and only differ in whether they can dynamically change their focus when generating each symbol.

### 2.2.3 LISTNER AGENTS

Listener tries to predict the target object from a set of candidate objects $C = \{o_1, o_2, ..., o_{|C|}\}$ given the speaker message $m$ by computing message-object matching scores $\{s_1, ..., s_{|C|}\}$ and choosing the object with the maximum score. Listener first encodes the objects using the object encoder and also encodes each symbol in the message into vectors $\{\mathbf{m}^1, ..., \mathbf{m}^T\}$ using a message encoder, for which the LSTM-based agent uses the bidirectional LSTM and the Transformer-based agent uses the Transformer encoder.

**Non-attention (NoAT) Listener** encodes each candidate object into a single vector $\hat{\mathbf{o}}_i$. The agent also averages the encoded symbol vectors into a single vector $\mathbf{m} = \frac{1}{T}\sum_{i=1}^{T}\mathbf{m}^i$. The message-object matching score is computed by taking the dot product of the object and message vector $s_i = \hat{\mathbf{o}}_i^{\top}\mathbf{m}$. In our experiments, the LSTM model uses the bidirectional LSTM encoder and the Transformer model uses the Transformer encoder.

**Attention (AT) Listener** encodes each object into a set of attribute vectors $\{\hat{\mathbf{o}}_i^1, ..., \hat{\mathbf{o}}_i^A\}$ and use the encoded symbol vectors as it is. With each encoded symbol vector $\mathbf{m}^t$ as query, the model produces an attention vector $\hat{\mathbf{m}}_i^t$ with the object attribute vectors as key-value using the dot-product attention. Intuitively, the attention vector $\hat{\mathbf{m}}_i^t$ is supposed to represent the attributes of the object $o_i$ relevant to the symbol $m^t$. Then, the symbol-object matching score is computed by taking the dot product between the attention vector and the message vector: $s_i^t = \hat{\mathbf{m}}_i^{t\top}\mathbf{m}^t$. Finally, the symbol-object matching scores are averaged to produce the message-object matching score: $s^i = \frac{1}{T}\sum_t s_i^t$.

## 2.3 OPTIMIZATION

The parameters of Speaker $\theta_S$ and Listener $\theta_L$ are both optimized toward the task success.

Speaker is trained with the REINFORCE algorithm (Williams, 1992). The message decoder produces the probability distribution of which symbol to generate $\pi_{\theta_S}(\cdot|t)$ at each time step $t$. At training time, message symbols are randomly sampled according to the predicted probabilities and the loss function for the Speaker message policy is $\mathcal{L}_\pi(\theta_S) = \sum_t r\log(\pi_{\theta_S}(m_t|t))$ where $m_t$ denotes the $t$-th symbol in the message. The reward $r$ is set to 1 for the task success and 0 otherwise.

As an auxiliary loss function, we employ an entropy regularization loss $L_H(\theta_S) = -\sum_t H(\pi_{\theta_S}(\cdot|t))$, where $H$ is the entropy of a probability distribution, to encourage exploration. We also add a KL loss $L_{\text{KL}}(\theta_S) = \sum_t D_{\text{KL}}(\pi_{\theta_S}(\cdot|t)\|\pi_{\bar{\theta}_S}(\cdot|t))$, where the policy $\pi_{\bar{\theta}_S}$ is obtained by taking an exponential moving average of the weights of $\theta_S$ over training, to stabilize the training (Chaabouni et al., 2022). In summary, the final speaker loss is $\mathcal{L}(\theta_S) = \mathcal{L}_\pi(\theta_S) + \alpha\mathcal{L}_H(\theta_S) + \beta\mathcal{L}_{KL}(\theta_S)$, where $\alpha$ and $\beta$ are hyperparameters.

Listener is trained with a multi-class classification loss. The message-object matching scores are converted through the softmax operation to $p_{\theta_L}(o_i|C)$, the probability of choosing the object $o_i$ as the target from the candidate set $C$. Then Listener is trained to maximize the probability of predicting the target object by minimizing the loss function $\mathcal{L}(\theta_L) = -\log p_{\theta_L}(o_{tgt}|C)$. The details of hyperparameter settings and training procedures can be found in Appendix A.3.

## 2.4 EVALUATION METRICS

We quantitatively evaluate emergent languages from how well the language can be used to solve the task and how well the language exhibits compositionality.

**Training accuracy (TrainAcc)** measures the task performance with objects seen during training. This indicates how the agent architectures are simply effective to solve the referential game.

**Generalization accuracy (GenAcc)** measures the task performance with objects unseen during training. We split the distinct object types in the game into train and evaluation sets and the generalization accuracy is computed with the evaluation set. As each object can be represented as a combination of attribute values, what we expect for the agents is to learn to combine symbols denoting each attribute value in a systematic way so that the language can express unseen combinations of known attribute values.

**Topographic similarity (TopSim)**, also known as Representational Similarity Analysis (Kriegeskorte et al., 2008), is one of the most commonly used metrics to assess the compositionality of emergent language (Brighton & Kirby, 2006; Lazaridou et al., 2018; Ren et al., 2020; Chaabouni et al., 2020). Intuitively, TopSim checks if similar objects have similar messages assigned. To compute TopSim, we enumerate all the object-message pairs $(o^1, m^1), ...., (o^{|\mathcal{O}|}, m^{|\mathcal{O}|})$ with a trained Speaker and define a distance function for objects $d_\mathcal{O}(o^i, o^j)$ and messages $d_\mathcal{M}(m^i, m^j)$. Then we compute Spearman's correlation between pairwise distances in the object and message space. For the distance function for objects $d_\mathcal{O}(o^i, o^j)$, we use the cosine distance of the binary attribute value vectors and for the distance function of messages $d_\mathcal{M}(m^i, m^j)$ the edit distance of message symbols.

## 3    EXPERIMENTAL SETUP

### 3.1    ONE-HOT GAME

Our first experimental setting is the referential game with objects consisting of one-hot vectors (Lazaridou et al., 2017; Kottur et al., 2017; Chaabouni et al., 2020; Guo et al., 2022). The objects are represented as a combination of two attributes (e.g., shape and color), which is a popular setting in the literature (Li & Bowling, 2019; Ren et al., 2020). Each attribute has 8 possible values, which results in 64 distinct objects. To evaluate generalization to unseen objects, we randomly split the 64 objects into training and evaluation sets at a ratio of 48/16. The number of candidates is set to 16, which is the maximum number of objects available in the evaluation set.

**Input Representation.** The objects are disentangled in terms of attributes and represented as a set of one-hot vectors where each dimension indicates a different value of a different attribute (i.e., two 16-dim one-hot vectors).

**Agent Configurations.** The vocabulary size of the agents is set to 16 and the message length of Speaker is 2. In this setting, a perfectly compositional language that scores the maximum score for all the compositionality metrics would assign different symbols to each attribute value, and symbols at each position consistently refer to a single attribute.

### 3.2    FASHION-MNIST GAME

Attention has been shown to be able to associate a symbol and a relevant region of an image to solve the task (Xu et al., 2015; Yang et al., 2016). To develop human-like emergent languages, one important question is whether the attention agents are able to develop a language such that we can understand the meaning of each symbol by inspecting the attended region. For this purpose, we design a multi-item image referential game using the Fashion-MNIST dataset[1] (Xiao et al., 2017).

In this game, each object is defined as a combination of two classes from the Fashion-MNIST dataset (e.g., *T-shirt* and *Sneaker*). The dataset contains 10 classes and thus the game has $_{10}C_2 = 45$ different types of objects. The 45 types are randomly divided into training and evaluation sets at a ratio of 30/15, and the number of candidates is set to 15.

**Input Representation.** Each object is presented to the agents as feature vectors extracted from a pixel image. The image is created by placing two item images, each of which is rescaled to the size of $48 \times 48$, on a $224 \times 224$ black canvas (Figure 4). The places of the items are randomly sampled so that the items never overlap.

The specific item images and their positions are randomly sampled every time the agents process the objects both during training and evaluation time to avoid degenerated solutions that exploit spurious features of an image (Lazaridou et al., 2018; Bouchacourt & Baroni, 2018). Also, the Speaker and Listener are presented with different images as the target object to facilitate learning a robust communication protocol (Rodríguez Luna et al., 2020).

Each image is encoded into $7 \times 7 \times 768$-dim feature vectors with a pretrained ConvNet[2] (Liu et al., 2022). For non-attention models, the feature vectors are averaged across spatial axes into a single 768-dim feature vector.

**Agent Configurations.** The vocabulary size of the agents is set to 20 and the message length is 2. A perfectly compositional language would refer to each item in the image with different symbols with a consistent one-to-one mapping.

## 4    RESULTS

### 4.1    ATTENTION AGENTS FIND MORE COMPOSITIONAL SOLUTIONS

We evaluate non-attention agents and attention agents where either/both Speaker and Listener have dynamic attention. We perform hyperparameter tuning as described in Appendix A and plot the

---

[1]https://github.com/zalandoresearch/fashion-mnist

[2]The pretrained model is registered as `convnext_tiny` in the `torchvision` library

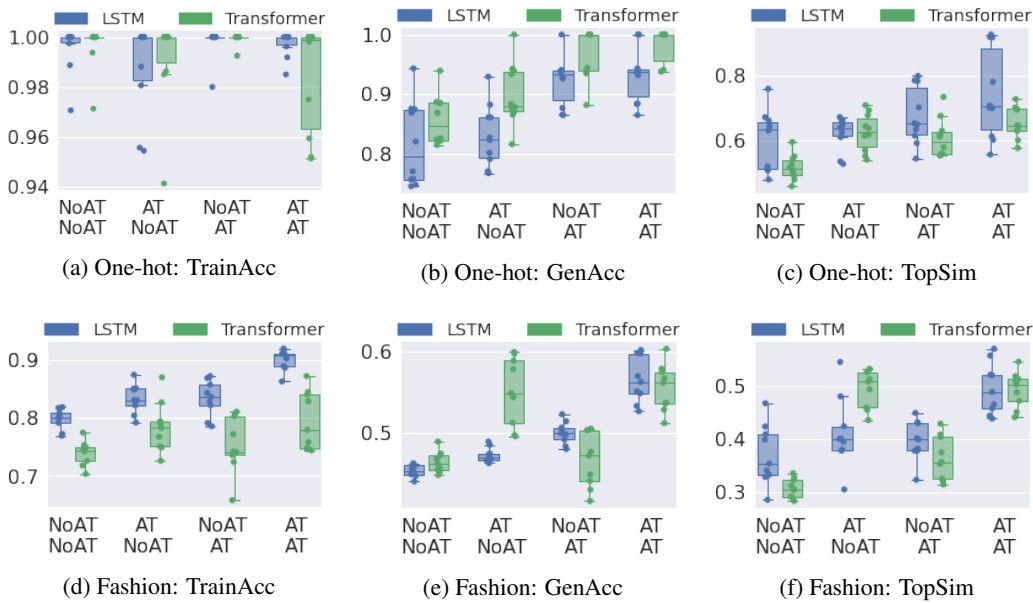

Figure 2: The results of the referential game. The color of the boxes indicates the base architecture of the agents (LSTM or Transformer) and the x-axis labels indicates whether Speaker and Listener use attention.

results of top-10 agents in terms of GenAcc for each setting on Figure 2. We do not discuss the difference between the LSTM and Transformer agents as they differ in multiple dimension in their architectures. Here, we focus on the difference between the non-attention and attention agents within each architecture.

We observe a general trend that the attention agents perform better than the non-attention baseline (NoAT-NoAT), which is indicated by the better average scores in task generalization (GenAcc) and compositionality metrics (TopSim) with different degrees of significance. This observation provides evidence in support of the hypothesis that the attention mechanism creates pressure for learning more compositional emergent languages.

One possible interpretation is that the attention mechanism adds flexibility to the model and it simply leads to better learning of the task. We observe the effect in the TrainAcc of the Fashion-MNIST game, where the attention models consistently outperform their non-attention baselines. However, we can still see the contribution of the attention mechanism besides the flexibility. We can see some NoAT-NoAT agents and AT-AT agents exhibit comparable TrainAcc around 75% in the Fashion-MNIST game, which means they are successful at optimization to a similar degree. However, we observe all the AT-AT agents significantly outperform any of the NoAT-NoAT agents, which indicates the degree of optimization alone cannot explain the better GenAcc and TopSim scores of the attention model. Therefore, the results supports our initial hypothesis that the attention mechanism facilitates developing compositional languages.

In some settings, we observe that adding the attention mechanism to only one of Speaker and Listener does not lead to better TopSim scores compared the NoAT-NoAT agents to as in the AT-NoAT LSTM agents in the one-hot game, and the AT-NoAT and NoAT-AT LSTM agents in the Fashion-MNIST game. However, when both Speaker and Listener agents have the attention mechanism, they all exhibit more generalizable and compositional languages, which indicates the effect of attention in Speaker and Listener is multiplicative.

## 4.2 ATTENTION AGENTS LEARN TO ASSOCIATE INPUT ATTRIBUTES AND SYMBOLS

Having confirmed that the attention agents give rise to more compositional languages, we proceed to examine if they use attention in an expected way, i.e., producing/understanding each symbol by associating them with a single input concept. We focus on analyzing the Transformer agents below.

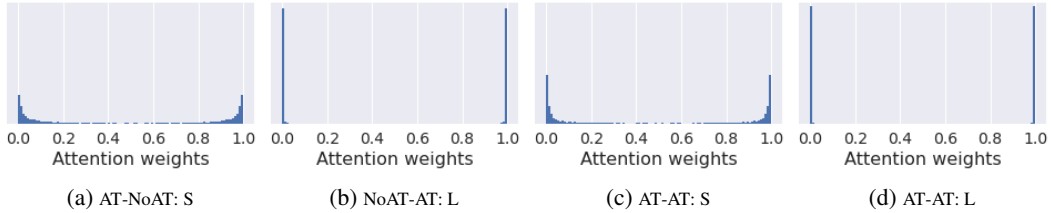

|  (a) AT-NoAT: S  |  (b) NoAT-AT: L  |  (c) AT-AT: S  |  (d) AT-AT: L  |

Figure 3: The distribution of attention weights from Transformer agents in the one-hot game.

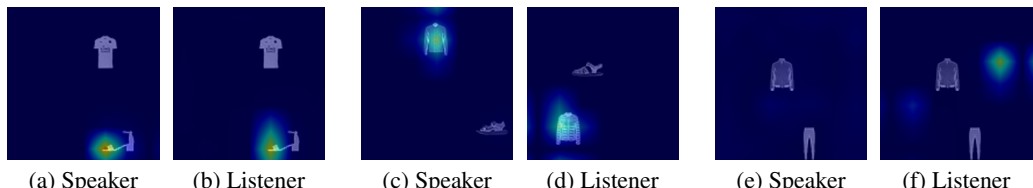

| (a) Speaker | (b) Listener | (c) Speaker | (d) Listener | (e) Speaker | (f) Listener |

Figure 4: Images with the attention maps produced by a AT-AT agent pair.

Inspecting the distributions of the attention weights assigned to each input attribute vector in the one-hot game, we observed that most attention weights are around either 0 or 1, indicating that the attention agents generally learn to focus attention on a single value (Figure 3). We also inspect the attention weights in the Fashion-MNIST game and confirm that attention agents generally learn to focus on a single object when generating a symbol as in Figure 4(a)-(d), although there are some failure cases as shown in Figure 4(e)-(f).

Given the observations above, we can associate each symbol in each message with the concepts defined in the game via attention weights. We visualize the association between each symbol in the vocabulary and each concept from a pair of AT-AT agents in Figure 5 to inspect the mapping patterns developed by the agents. A symbol is considered to be associated with a concept when the attribute value has the largest attention weight in the one-hot game or the center of gravity of the attention weights is within the bounding box of the item in the Fashion-MNIST game. Overall, we observe that the mappings learned by Speaker and Listener are mostly aligned except for the symbol c in the one-hot game and the symbol i in the Fashion-MNIST game. We identify three types of symbol-to-concept mapping patterns.

**Monosemy.** A single symbol always refers to a single concept, e.g., a, b, and d in the one-hot game; a, e, and g in the Fashion-MNIST game. This is a desired mapping pattern that allows an unambiguous interpretation of symbols. However, we also observe "synonyms" where a single concept is referred to by multiple symbols, e.g., 1-3 by d and e in the one-hot game.

**Polysemy.** A single symbol refers to multiple concepts, e.g., j-n in the one-hot game; b, c, and d in the Fashion-MNIST game. These symbols are somewhat ambiguous, but it is likely that the agents develop a way to resolve the ambiguity, especially in the one-hot game where agents achieve more than 90% average accuracy scores with unseen objects. Moreover, these polysemous symbols always refer to two values of the different attributes. Given these observations, the agents are likely to have learned a consistent word order in terms of the object attributes and disambiguated the polysemous symbols by their position.

The polysemous symbols in the Fashion-MNIST game seem to be affected by the visual similarity of the fashion items, e.g., b refers to tops (Pullover, Coat, and Shirt) and f refers to shoes (Sandal, Sneaker, and Boot). This is in contrast to the one-hot game, where the object attribute representations are initialized randomly, which probably led to an arbitrary assignment of polysemy. The polysemous patterns in the Fashion-MNIST game demonstrate that the semantics of emergent language can be heavily influenced by the property of the input objects.

**Gibberish.** We observe a few cases where the attention weights do not consistently focus on any particular regions in the Fashion-MNIST game, e.g., the symbol i. These gibberish symbols could have conveyed something informative but uninterpretable to humans, but we confirmed that the

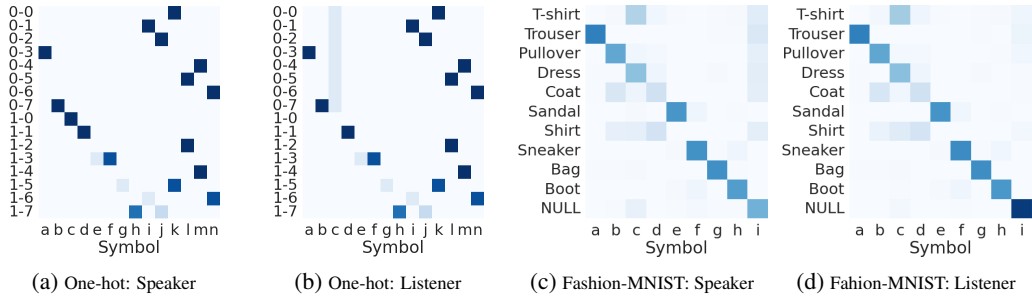

(a) One-hot: Speaker  (b) One-hot: Listener  (c) Fashion-MNIST: Speaker  (d) Fahion-MNIST: Listener

Figure 5: The mappings between symbols and concepts derived from the attention weights of Transformer AT-AT agent pairs. The darkness correslates with the frequency of the association.

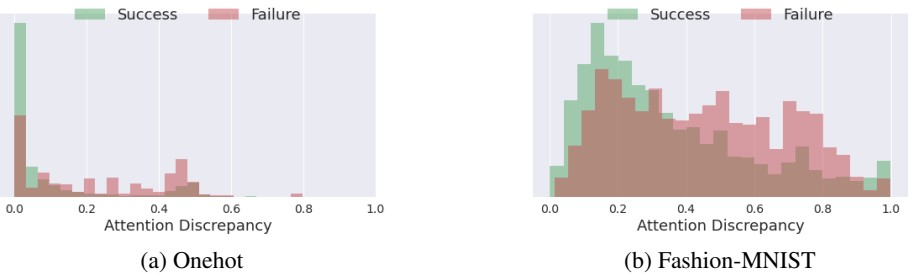

(a) Onehot  (b) Fashion-MNIST

Figure 6: The distribution of attention discrepancy scores from the trials of the AT-AT agents. The frequency is normalized within successful and failed trials respectively.

communication success rate with the gibberish symbols is much lower (19.4%) than the overall success rate (45.3%), indicating that the gibberish symbols are the results of optimization failure.

### 4.3 THE COORDINATION OF SPEAKER AND LISTENER ATTENTIONS PREDICTS THE TASK PERFORMANCE

An important prerequisite of successful communication is the participants engaging in joint attention and establishing a shared understanding of each word (Garrod & Pickering, 2004). Here we show that the degree of the alignment between Speaker's and Listener's attention weights can be regarded as a proxy of their mutual understanding and predictive of communication success.

We define a metric called *attention discrepancy*, which measures the difference between the attention weights of Speaker and Listener given the same inputs. The metric is calculated by computing the Jensen–Shannon Divergence (Lin, 1991) between the Speaker's and Listener's attention maps.

Figure 6 shows the frequency of attention discrepancy scores within successful and failed communications.[3] We can see that successful communications can be characterized by lower attention discrepancy compared to failed communications, indicating that the alignment of the attention weights does correlate with communication success. This suggests that the attention agents largely develop intuitive communication protocols that require a shared understanding of symbols.

## 5 RELATED WORK

### 5.1 EMERGENT LANGUAGE

One of the desired properties of emergent language is *compositionality* because more compositional languages are supposed to be intuitive, interpretable, generalizable, and easier to learn (Li & Bowl-

---

[3]We observe small peaks around 0.5 in the one-hot game for both successful and failed communications, and at a closer look, this corresponds to cases where the Speaker and Listener attend to a different attribute at one of the two symbols in the message.

ing, 2019; Ren et al., 2020). Compositional languages in this context usually have symbols each of which corresponds to one of the primitive concepts in the environment and combine them in a straightforward manner to form a message. Existing studies have investigated various environmental pressures to improve the compositionality of emergent language: learning across generations (Li & Bowling, 2019; Ren et al., 2020), learning with a population (Rita et al., 2022), applying noise to communication channel (Kuciński et al., 2021).

These recent studies have been mainly conducted with simple RNN-based Speaker and Listener agents, where the inputs (e.g., attribute values or raw images) are encoded into a single input vector for subsequence computations. Other agent architectures have been employed for other specific motivations. Chaabouni et al. (2019) and Ryo et al. (2022) used the LSTM sequence-to-sequence (with attention) architecture to study how agents transduce a sequential input generated from a certain grammar to their own emergent language. Gupta et al. (2021) developed a patch-based Speaker architecture that produces messages by focusing important image patches.

One of our goals in this study is to investigate how the inductive biases of the Speaker and Listener architectures affect the compositionality of the emergent language. In a similar spirit, Słowik et al. (2020) compared Speaker agents which process inputs in the form of a graph, sequence, and bag-of-words and show that the graph architecture exhibits more compositional emergent languages. Our study offers additional empirical evidence in this direction with the attention mechanism.

## 5.2 ATTENTION MECHANISM

The attention mechanism has been shown very effective in supervised learning (Xu et al., 2015; Vaswani et al., 2017) and has become an indispensable modeling piece in modern neural networks. Conceptually, the attention mechanism models pairwise associations between a query vector and a subset of key-value vectors. This enables the model to focus on a subpart of compositional representation and has been shown to enforce composition solutions in various settings such as visual reasoning (Hudson & Manning, 2018), symbolic reasoning Korrel et al. (2019), image generation (Hudson & Zitnick, 2021). Our study demonstrates that by modeling association between a symbol and an object attribute with attention, the agents can find more compositional languages in the referential game without any additional supervision.

Another potential benefit of attention is interpretability. Attention has been shown to provide plausible alignment patterns between inputs and outputs, for example, source and target words in machine translation (Bahdanau et al., 2015) and image regions and words in image captioning (Xu et al., 2015; Yang et al., 2016). On the other hand, the alignment patterns may need careful interpretation. The patterns are not always consistent with human intuitions (Alkhouli et al., 2018; Liu et al., 2020) and there is an ongoing debate about to what extent the attention weights can be used as explanations for model prediction (Jain & Wallace, 2019; Wiegreffe & Pinter, 2019; Bibal et al., 2022). In this paper, our agents show relatively straightforward attention patterns that allow unambiguous interpretations but the interpretability of attention weights with more complex models and environments needs further investigation in the future.

## 6 DISCUSSION AND CONCLUSION

When we face a complex object, we are likely to dynamically change our focus on its subpart to describe the whole (Rensink, 2000). Hearing a word helps us quickly recognize its referent (Boutonnet & Lupyan, 2015) and affects where to focus in an image (Estes et al., 2008). Motivated by these psychological observations, we implemented agents with the dynamic interaction between symbols and input representations in the form of the attention mechanism. We showed that the attention agents develop more compositional languages than their non-attention counterparts. This implies that the human ability to focus might have contributed to developing compositional language.

Overall, our results suggest that future work should explore more architectural variations that originate from human cognitive processing to gain insights into the effect of the cognitive property on shaping a language. Given the observation that the alignment of Speaker's and Listener's attention is somewhat indicative of successful communication in §4.3, for example, future work can explore how incorporating joint attention (Kwisthout et al., 2008) into training helps to learn a language.

## REPRODUCIBILITY STATEMENT

The experiments described in this paper are ensured to be reproducible in the following ways:

- We describe the task setup and models in §2 together with more specific descriptions of each task in §3. The additional details and hyperparameters to reproduce the results are given in appendix A.
- The artifacts used in the experiments (the Fashion-MNIST dataset and pretrained ConvNet encoder in §3.2) are all publicly available.
- The source code will be released upon acceptance.

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

APPENDIX

## A  ADDITIONAL DETAILS ON EXPERIMENTAL SETUP

### A.1  MINI-BATCH CREATION

Each mini-batch in our referential game is specified via two parameters: the batch size $b$ and candidate size $|C|$. We first sample $b$ objects from the set of the objects available in the game $O$. Each object in a mini-batch serves as the target object in a round of the game, and we sample $|C| - 1$ distractors for each target object. Importantly, we sample the objects so that every object in a round of the game has a different object type.

### A.2  DATA PROCESSING IN THE FASHION-MNIST GAME

The images used in the Fashion-MNIST game (§3.2) are created by sampling item images from the Fashion-MNIST dataset (Xiao et al., 2017). We first take 5,000 images for each item class in the dataset, and then create image objects for the referential game by sampling from the image pool. Instead of generating the image and input features on the fly during training, we precompute 3,000 images and their features for each object class (i.e., each combination of two items), and they are randomly sampled during training and evaluation.

### A.3  HYPERPARAMETERS

The training setups and hyperparameters are shown in Table 1. The hyperparameter tuning is conducted for the weight of entropy loss $\alpha$, and the scores in the paper are computed from the agents with the best 10 generalization accuracy score for each agent.

|  | One-hot (4, 4) | Fashion-MNIST |
|---|---|---|
| Speaker message embedding dim | 64 | 256 |
| Speaker LSTM decoder hideen dim | 64 | 256 |
| Speaker Transformer decoder dim | 64 | 256 |
| Speaker Transformer decoder feed-forward dim | 128 | 512 |
| Listener message embedding dim | 64 | 256 |
| Listener Bi-LSTM encoder hidden dim | 32 | 128 |
| Listener Transformer encoder dim | 64 | 256 |
| Listener Transformer encoder feed-forward dim | 128 | 512 |
| Training Batch size | 48 | 480 |
| Max Training steps | 15K | 50K |
| Evaluation Rounds | 1500 | 15000 |
| Entropy loss weight $\alpha$ | [0.1, 0.01, 0.001] | [0.1, 0.01, 0.001] |
| KL loss weight $\beta$ | 0.1 | 0.1 |
| Learning rate | 1e-4 | 1e-4 |

Table 1: Hyper-parameters of the experiments.

## B  ADDITIONAL RESULTS

For the configuration of the one-hot game in the main experiments, the number of attributes is 2 and the number of the possible values is 8. We denote this game setting as the one-hot game (8, 2). Here, we show results from the one-hot games with different configurations with more attributes (4, 4) and more values (16, 2). The capacity of the communication channel is set to $|V| = 16, T = 4$ and $|V| = 32, T = 2$ respectively. Also, to accommodate the increased task complexity, we double the model size (i.e., every hidden vector dimension) and set the training batch size to 192 and the max training steps to 30K.

The results are summarized in Figure 7. We generally observe the same trend as discussed in §4.1: AT-AT agents outperform their corresponding AT-AT baselines in GenAcc and TopSim. The only

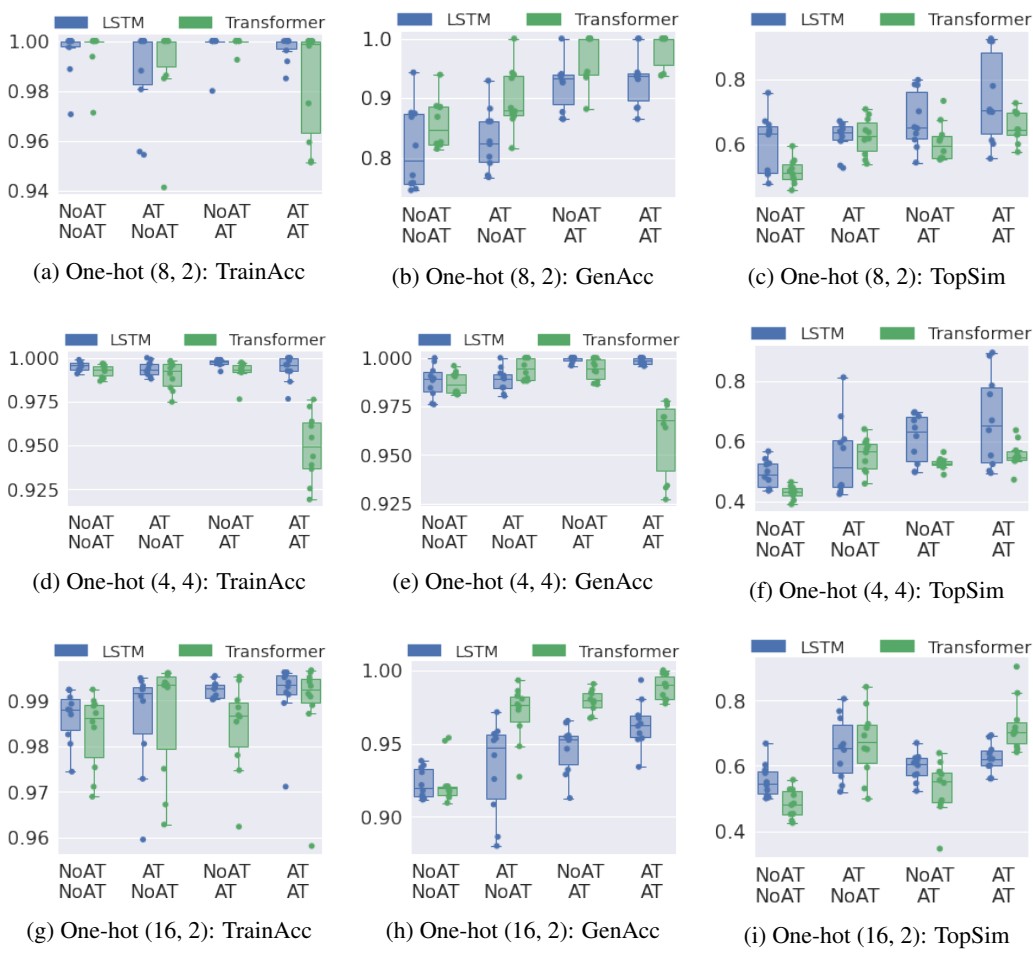

Figure 7: The results of the one-hot game with different configurations. The color of the boxes indicates the base architecture of the agents (LSTM or Transformer) and the x-axis labels indicates whether Speaker and Listener use attention.

exception is GenAcc of the AT-AT Transformer agents in the one-hot (4, 4) game. The AT-AT agents show significantly lower GenAcc average score than NoAT-NoAT agents (Figure 7e) as well as the TrainAcc score (Figure 7d). We think that this is caused by the instability of jointly coordinating attention between Speaker and Listener during training. This can be particularly problematic with a long message length. Yet, the average TopSim score is still higher than the NoAT-NoAT agents (Figure 7f), indicating the inductive bias towards more compositional languages is still effective in this case.

## C   THE DETAILS OF THE SPEAKER ARCHITECTURES

In this section, we provide detailed descriptions of the Speaker agents used in the paper.

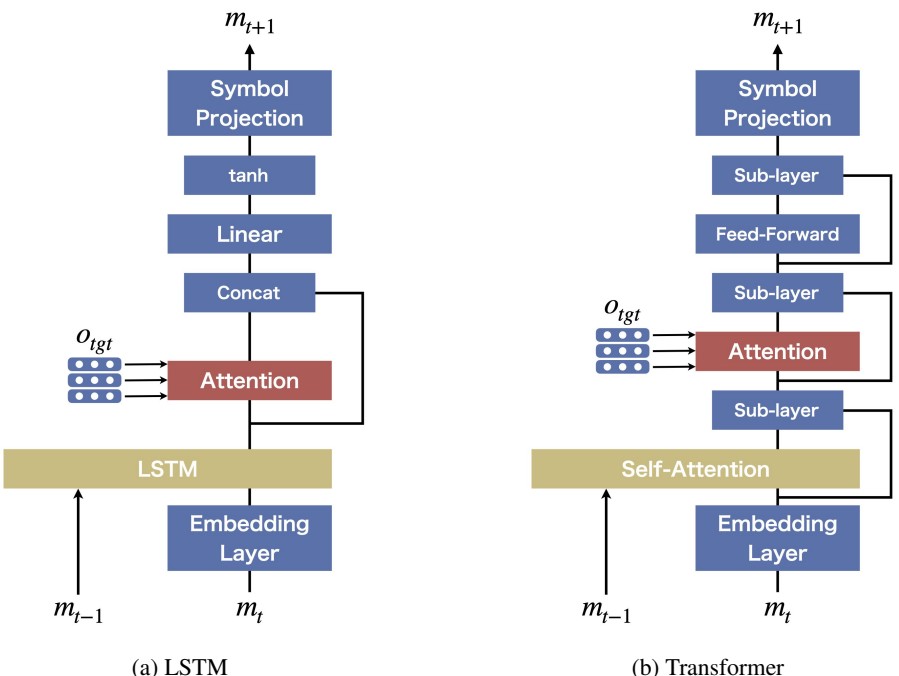

(a) LSTM                                        (b) Transformer

Figure 8: The architecture of the LSTM and Transformer decoders used in this paper.

### C.1   SPEAKER

Speaker has an auto-regressive message decoder that takes the encoded vector(s) of the target object $\{\hat{\mathbf{o}}^1, ..., \hat{\mathbf{o}}^A\}$ as input and generates a multi-symbol message $m = (m_1, ..., m_T)$.

At each time step $t$, the decoders embed a previously generated symbol into a vector $\mathbf{m}_{t-1}$ and produce an output hidden vector $\mathbf{h}_t$, which is fed into the output projection layer to predict the next symbol $m_t$. The LSTM and Transformer decoders differ in how they transform the symbol vector $\mathbf{m}_{t-1}$ into the output hidden vector $\mathbf{h}_t$. We decompose the internal computation into three steps (1) recurrent computation; (2) attention; (3) post-processing, and how each step is implemented.

#### C.1.1   LSTM

The LSTM decoder is based on the global attention decoder introduced in Luong et al. (2015).

**Recurrent computation.** The LSTM decoder updates the input symbol vector $\mathbf{m}_{t-1}$ with the contextual information using the LSTM cell (Hochreiter & Schmidhuber, 1997). We initialize the memory cell vector with the average of the object vectors $\frac{1}{A}\sum_{i=1}^{A}\hat{\mathbf{o}}^i$ followed by a linear transformation and the hidden vector with zero. We also apply layer normalization (Ba et al., 2016) to the hidden vector at each time step.

**Attention.** With the output of the LSTM cell $\mathbf{x}_t$, the attention vector is computed over the object vectors $\{\hat{\mathbf{o}}^1, ..., \hat{\mathbf{o}}^A\}$ using the bilinear attention. The attention score for each object attribute $o^i$ is computed as $s_t^i = \mathbf{x}_t^\top \mathbf{W}_b \hat{\mathbf{o}}^i$, where $\mathbf{W}_b$ is a learnable matrix. Then the scores are used to produce the attention weights via the softmax function $a_t^i = \exp(s_t^i)/\sum_j \exp(s_t^j)$. Finally, the object vectors are used as the value vectors to produce the output attention vector $\hat{\mathbf{x}}_t = \sum_{i=1}^{A} a_t^i \hat{\mathbf{o}}^i$.

**Post-processing.** Finally, the attention vector $\hat{\mathbf{x}}_t$ is concatenated with $\mathbf{x}_t$ and the vector is projected into the dimension of the object vector followed by the $\mathrm{tanh}$ activation function: $\mathbf{h}_t = \mathrm{tanh}(\mathbf{W}_o[\hat{\mathbf{x}}_t; \mathbf{x}_t])$, where $\mathbf{W}_o$ is a learnable matrix.

### C.1.2 TRANSFORMER

The Transformer decoder is based on the original architecture introduced in Vaswani et al. (2017). We first explain some building blocks repeatedly used in the computation.

**Multi-head QKV Attention.** The attention used in Transformer takes the query vector $\mathbf{x}$ and key-value vectors $\{\mathbf{y}_1, ..., \mathbf{y}_L\}$ as input and is based on the QKV attention. It first project the vectors into query, key, value vector spaces respectively by a linear transformation: $\mathbf{q} = \mathbf{W}_q\mathbf{x}, \mathbf{k} = \mathbf{W}_k\mathbf{y}, \mathbf{v} = \mathbf{W}_v\mathbf{y}$, where $\mathbf{W}_q, \mathbf{W}_k, \mathbf{W}_v$ are learnable matrices. Then the attention score is calculated using the scaled dot attention: $s_t^i = (\mathbf{q}_t^\top \mathbf{k}^i)/\sqrt{d}$, where $d$ is the dimension of the query and key vectors. The scores are transformed into the attention weights via the softmax function $a_t^i = \exp(s_t^i)/\sum_j \exp(s_t^j)$. Finally, the attention vector is calculated as the weighted sum of the value vectors: $\hat{\mathbf{x}}_t = \sum_{i=1}^A a_t^i \mathbf{v}^i$.

Transformer further combines the QKV attention with the multi-head mechanism. In essence, it divides the query, key and value vectors into sub-spaces with the same dimension, and perform the attention computation as above within each sub-space, and then concatenate each output. We denote the attention computation as $\hat{\mathbf{x}} = \mathrm{MultiHeadQKV}(\mathbf{x}, \{\mathbf{y}_1, ..., \mathbf{y}_L\})$ below.

**Sub-layer connection.** The Transformer architecture combats the gradient vanishing problem using sub-layer connection after each module. Let the input or hidden vector of the current time step be $\mathbf{x}$ and a module to update the vector as Module, and the sub-layer connection can be described as: $\hat{\mathbf{x}} = \mathrm{SubLayer}(\mathbf{x}, \mathrm{Module}(\cdot)) = \mathrm{LN}(\mathrm{Module}(\mathbf{x}) + \mathbf{x})$, where LN denotes layer normalization.

**Recurrent computation.** The Transformer decoder updates the input symbol vector $\mathbf{m}_{t-1}$ with the contextual information using the self-attention mechanism. The vector $\mathbf{m}_{t-1}$ attends to the previous inputs with the multi-head attention mechanism and produce the updated vector: $\mathbf{x}_t' = \mathrm{MultiHeadQKV}(\mathbf{m}_{t-1}, \{\mathbf{m}_1, ..., \mathbf{m}_{t-1}\})$. Then the sub-layer connection is applied to produce $\mathbf{x}_t = \mathrm{LN}(\mathbf{x}_t' + \mathbf{m}_{t-1})$.

**Attention.** The updated vector $\mathbf{x}_t$ is further updated by attending to the object vectors $\{\hat{\mathbf{o}}^1, ..., \hat{\mathbf{o}}^A\}$: $\hat{\mathbf{x}}_t' = \mathrm{MultiHeadQKV}(\mathbf{x}_t, \{\hat{\mathbf{o}}^1, ..., \mathbf{o}^A\})$, followed by the sub-layer connection: $\hat{\mathbf{x}}_t = \mathrm{LN}(\hat{\mathbf{x}}_t' + \mathbf{x}_t)$.

**Post-processing.** Finally, the attended vector $\hat{\mathbf{x}}_t$ undergoes the feed-forward layer: $\mathbf{h}_t' = W_2\mathrm{ReLU}(W_1\hat{\mathbf{x}}_t)$, where $W_1$ and $W_2$ are learnable matrices. The dimension of $\mathbf{h}_t'$ should be the same as the input $\hat{\mathbf{x}}_t$ to apply the sub-layer connection: $\mathbf{h}_t = \mathrm{LN}(\hat{\mathbf{h}}_t' + \hat{\mathbf{x}}_t)$.

# D ANOTHER POSSIBLE BASELINE

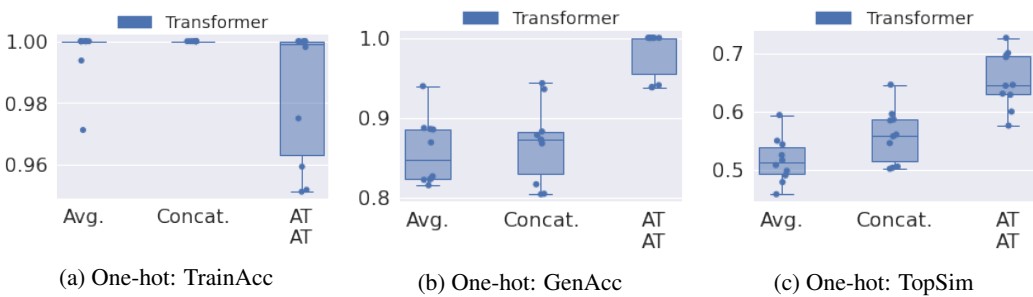

(a) One-hot: TrainAcc    (b) One-hot: GenAcc    (c) One-hot: TopSim

Figure 9: The comparison of different types of baseline object encoders.

In the main experiments, we used the object encoder that simply averages the attribute vectors for the non-attention baseline, which is equivalent to the model attending to the attribute vectors uniformly (i.e., with the same attention weights). Another possible baseline that aggregates multiple attribute vectors into one fixed-sized vector can incorporate the concatenation of the attribute vectors. Such method creates an object vector where the attribute information is disentangled across the dimension and thus may lead to better compositionality in the emergent language.

Here, we replace the object encoder of the NoAT-NoAT agent in the one-hot game with the concatenation-based one. It firstly projects the input attribute vectors $\{\mathbf{o}^1, ..., \mathbf{o}^A\}$ into the dimension of $d/A$, where $d$ is the dimension of the final object vector, with a learnable matrix parameter. Then, the final object vector $\hat{\mathbf{o}}$ is derived by concatenating the projected vectors.

Figure 9 shows the result from the average-based and concatenation-based object encoder. We observe that the concatenation-based encoder outperforms the average-based encoder in TopSim, indicating that having the attribute information disentangled across the vector dimension facilitates learning compositional languages. Still, the AT-AT agents perform the best, which shows that having dynamic attention at each symbol generation is more effective.

## E  THE EFFECT OF MULTI-HEAD ATTENTION

In the main experiments, we used a single-head attention for the Transformer decoder to facilitate a simpler analysis. In this section, we check whether using the multi-head attention affects the performance of the agents. We modify the number of heads in the source-target attention of the attentional Transformer Speaker and experiment with the non-attention Listener in the one-hot game. We could expect that the multi-head attention may improve the performance by adding more flexibility in learning or to the contrary hurt the performance to lose the inductive bias of one-to-one mapping between a symbol and an object attribute.

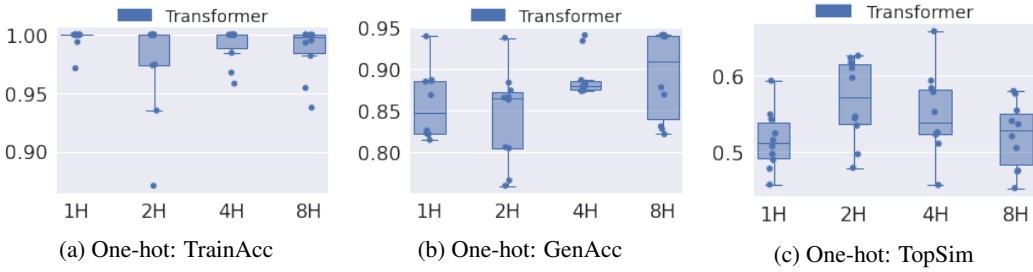

(a) One-hot: TrainAcc      (b) One-hot: GenAcc      (c) One-hot: TopSim

Figure 10: The comparison of different numbers of heads in the source-target attention of the Transformer Speaker. Listener is the non-attention Transformer model.

From Figure 10, changing the number of heads basically does not affect the performance except for the two-head attention, where GenAcc is significantly lower than the singe-head agent ($p < 0.01$) but the TopSim is significantly higher ($p < 0.01$). This possibly stems from some optimization instability. Overall, we do not observe significant impact of the multi-head attention in this case.

