# OpenReview forum: "Emergent Communication with Attention"
_ICLR.cc/2023/Conference — Submitted to ICLR 2023_

### Official Review · Reviewer_kyNy · 2022-10-24

**Confidence:** 3
**Correctness:** 3
**Technical Novelty And Significance:** 2
**Empirical Novelty And Significance:** 3
**Recommendation:** 5

**Clarity, Quality, Novelty And Reproducibility:**

The paper is generally clear, although some of the arguments could have been more developed in my view. For example, it is not made clear why compositionality is a relevant property *in this case* (I agree that this is an important property of language in general). Why should it emerge at all in this experimental setup: As I understand only identity is to be conveyed, but no attribute.

The novelty of the paper resides in its scientific question, namely that transformer-like attention mechanism induces more compositionality in emergent language. The technical content of the paper is well known and studied. I cannot comment on the novelty of this question, as I do not have an in-depth knowledge of compositionality in language.

The authors make clear that the source code will be made available on acceptance. Although all data is drawn from the publically-available Fashion-MNIST it would be important that the code to generate the exact same compositions of garments would be provided.

p.4 positioanl should be positional

**Strength And Weaknesses:**

### Strengths
- The paper is generally clear and pleasant to read.
- The question of language emergence and compositionality of codes in particular is an interesting and relevant one.
- The experimental setup is well described and clear.

### Weaknesses
- It remain unclear to me whether the chosen experimental design is well suited to the research question.
	- First, the FashionMNIST scenario is only based on identifying pairs of items, there are no properties attached to the items, so it is not clear what type of compositionality we should expect.
	- Second, the message length is 2, therefore it is unclear how compositionality really comes into play. The argument is made on the pre-message dimensions of the attention model, but is that really the language in this experiment? This could be made clearer.
- The attention mechanism used in transformers, if very successful in neural networks, are a very specific form of attention, there are other. This is not conveyed in the related work section. I would have liked some more discussion on the specific properties of this type of attention and why it is particularly suited for the emergence of compositionality.
- The discussion of the results could be clearer: The results seem to show that attention increases compositionality (according to the chosen metrics), but also that attention generally improves performance at the task across the board. Is it the case that increased attention and therefore the benefit of attention is due to increased compositionality, or does compositionality increases due to better performance? I am not sure this question is answered by the experimental results.

**Summary Of The Paper:**

This paper is concerned with emergent communication: How pairs of agents (neural networks in this case), develop joint symbols for communicating information, and is in particular concerned with the compositionality of those symbols.

The hypothesis of this paper is that the mechanism of attention, similar to the one popularised in transformers, lead to more compositionality in the emergent language.

The experimental setup in this paper is based on the "referential game", where one agent("speaker") sees an object and need to convey this to the second ("listener"). The listener must then pick the correct object from a set of distractors. The speaker is implemented as a pre-trained CNN, the listener as a Transformer decoder. Experiments are run in a simple one-hot game and on inputs based on pairs of garments taken from the fashion MNIST dataset.

The experimental results are:
- the use of attention lead to better performance in the game
- the use of attention lead to more compositionality in the emergent language

**Summary Of The Review:**

In summary, this is a well written paper, tackling an interesting and important question, namely compositionality in emergent language.

The paper tries to establish the effect/importance of transformer-like attention in compositionality, but it is unclear to me that the results really answer this question, beyond showing the expected result that attention improves performance.

I have some concerns that the task may be too simple to effectively study the emergence of compositionality, as it may indeed not be needed at all.

The paper would also gain from a more in-depth discussion of the different mechanisms and approaches to attention and rationales why they would affect the emergence of compositionality.

---

> ### Author Response · Authors · 2022-11-18
> **First Response to Reviewer kyNy**
>
> > The paper is generally clear, although some of the arguments could have been more developed in my view.
> > For example, it is not made clear why compositionality is a relevant property in this case (I agree that this is an important property of language in general). Why should it emerge at all in this experimental setup: As I understand only identity is to be conveyed, but no attribute.
>
> Thank you for your comment.
> The part "only identity is to be conveyed" is actually important in the research of emergent communication (EC).
> One of the goals of the EC research is that clarifying the condition where compositionality "emerges" when it is not necessarily needed.
> In the referential game, the agents are only asked to convey the identity of objects and it is totally up to them what kind of encoding they use.
> They can be either non-compositional or compositional.
> As the compositionality is what human language has, the question here is how we can observe the agents develop compositional languages in the game.
>
> We have updated the description of the referential game in the newer version to make the motivation clearer.
>
> > The paper tries to establish the effect/importance of transformer-like attention in compositionality, but it is unclear to me that the results really answer this question, beyond showing the expected result that attention improves performance.
>
> Indeed this is a common concern raised by other reviewers as well.
> We would like to refer to our response to Reviewer 4TD5 on the interpretation of the results.
>
>
> > p.4 positioanl should be positional
>
> Thank you. We have fixed the typo accordingly.

---

### Official Review · Reviewer_gZp6 · 2022-10-24

**Confidence:** 5
**Correctness:** 3
**Technical Novelty And Significance:** 2
**Empirical Novelty And Significance:** 2
**Recommendation:** 3

**Clarity, Quality, Novelty And Reproducibility:**

The paper is clear and well-written. Actually, I would encourage the authors to make it more concise. It would allow more space to perform more experiments and ablation to nail the research questions.
The experiments are interesting but tend to be too narrow. In Emecom, a simple change of hyperparameters may drastically change results, and there were no attempts to try to refute the outcome. Besides,
More importantly, the paper focuses on model architecture... but never fully describes the model itself (only mentioning  Vaswani2017 is not enough when the core topic of the paper is transformer). Therefore, it greatly lowers paper reproducibility.
Similarly, a few details are missing on the training (number of images, batch size, etc.)

Small remarks:
- The fashion task description is not very clear. I would also refer to [1] to illustrate the task too
- Figure 5 is nice! This was really an excellent idea. To ease readability, I would recommend plotting cumulative normalized bars instead of overlapping bars

Finally, I would recommend the authors also include some work on transformer compositionality in natural language or images. Quick search provides many references [2-4]

[1] https://proceedings.neurips.cc/paper/2021/file/9597353e41e6957b5e7aa79214fcb256-Paper.pdf
[2] https://aclanthology.org/2022.acl-long.251.pdf
[3] https://vigilworkshop.github.io/static/papers-2019/43.pdf
[4] https://arxiv.org/abs/2111.08960


**Strength And Weaknesses:**

First of all, the authors point out a relevant fact: the language emergent models mostly remain stuck to RNN, while transformers were a game-changer in the Language modeling community. Therefore, updating the EmeCom modeling with transformers is worth exploring.
However, I think that the paper only scratches the surface of the problem:
 - The transformer is 1-attention layers, which are merely basic memory networks. In this case, I would also add an RNN with attention.
 - there is no actual discussions about the difference transformer impact on language (depth, encoder/decoder, decoder-only). This would be by far the most exciting part for me
 - the task is extremely toyish, while transformers shine when complexity increase. For instance, a message length of 2 prevents advanced compositionality,
 - there is no clear disentanglement between the vision-transformer and language transformer in the paper, while compositionality may come from different input

As a result, my main takeaway from the paper is that the transformer (single-attention layers) outperforms RNN-based system. Yet, even this point must be taken with caution due to the lack of extensive architecture search.

For instance, a simple idea to increase paper generability: Given different X tasks of increasing difficulty (few attributes with many values or vice-versa) or with different settings (small/long message length), I would explore different combinations of neural architecture (RNN, RNN+attention, different transformers) where network capacity is fixed. Then, if an attention-based network consistently outperform RNN based model (acc/topsim), then it is an interesting outcome.

**Summary Of The Paper:**

The paper explores how attention may impact language emergence in two instances of the Lewis Game setting.
The authors observe that transformers seem to introduce an inductive bias toward compositionality.
They then look at the alignment between concepts and attention weights.


**Summary Of The Review:**

While I value the initial research direction and some effort in analyzing the results (e.g., Fig5), the paper does not fully explore how transformers may impact the emergent language. I would recommend iterating over diverse tasks and architectures to discover potential correlations. Note that this does not require extensive computation if the network remains below a few million parameters on a simple one-hot or visual task.
Furthermore, I am missing reproducibility points.
Therefore, I cannot recommend paper acceptance in its current state. Yet, I recommend the authors to continue their research work.

---

> ### Author Response · Authors · 2022-11-18
> **First Response to Reviewer gZp6**
>
> > the paper does not fully explore how transformers may impact the emergent language. I would recommend iterating over diverse tasks and architectures to discover potential correlations.
>
> Thank you for your comment.
> The focus of our work is proving that the inductive bias from the attention mechanism improves compositionality rather than exploring Transformer.
> Our initial motivation to experiment with Transformer was that the architecture naturally incorporates the attention mechanism and it is widely used in the current ML field.
> However, as you mentioned, the experiments presented in the previous version lack generality in terms of task settings and model architectures.
> To remedy this, we have added the experimental results from the LSTM-based models, comparing non-attention and attention variants within each architecture.
> We also added results from the one-hot game with different configurations in Appendix B.
>
>
> > In Emecom, a simple change of hyperparameters may drastically change results, and there were no attempts to try to refute the outcome.
>
> We agree that we need to try more settings to validate the generality of the observation.
> We have added experimental results from the one-hot games with different configurations in Appendix B.
> We will try other configurations as well in the future.
>
>
> > the paper focuses on model architecture... but never fully describes the model itself
> > The fashion task description is not very clear. I would also refer to [1] to illustrate the task too
>
> Thank you for pointing out this.
> We have revised the description of the model architecture (Section 2) and task description (Section 3).
> For details we omit due to the space limit, we refer the readers to Appendix C.
>
> > Finally, I would recommend the authors also include some work on transformer compositionality in natural language or images. Quick search provides many references [2-4]
>
> Thank you for suggesting relevant work.
> Although we cannot include all of them due to the page limit, we have revised the related work section to give more focus on the idea of enforcing compositional solutions by using attention, and added some relevant papers including (Hudson and Zitnick, 2021) you suggested.
>
>
> > Figure 5 is nice! This was really an excellent idea. To ease readability, I would recommend plotting cumulative normalized bars instead of overlapping bars
>
> Thank you for your suggestion.
> Plotting cumulative normalized bars would improve the visibility, but we are afraid that it may hinder intuitive understanding of the graph, as the x-axis (attention discrepancy) does not have unidirectionality, which fits well with the cumulative graph, per se (e.g., "bad" <- -> "good", as opposed to "past" -> "future").
> We have kept the style, but feel free to point out if we misunderstood the suggestion.

---

### Official Review · Reviewer_4G68 · 2022-10-25

**Confidence:** 4
**Correctness:** 3
**Technical Novelty And Significance:** 2
**Empirical Novelty And Significance:** 3
**Recommendation:** 5

**Clarity, Quality, Novelty And Reproducibility:**

* Clarity: the text is generally clear;
* Novelty: I believe the findings are novel;
* Quality: the experiments seem to support the claims;
* Reproducibility: the authors state that the code would be open sourced upon acceptance.


**Strength And Weaknesses:**

Strengths:
* Generally the paper is well-written and has a good literature review.
* The results (Table 1) show a significant improvement over the baseline and the interpretability section (section 4.3) is interesting, even if the difference between the failure and success settings is not significant.

Weaknesses:
* The main novelty of this paper is agents’ architecture (section 2.2). However, this attention mechanism is not sufficiently explained.
The authors state that the query is the symbol of the message and the keys/values are the inputs’ representation. This is still a vague description of the mechanism.
* The baseline architecture simply averages input embeddings. Is this the most fair baseline we can come up with? For instance, can we have an architecture where the set of vectors are projected to a smaller dimension (/2) and concatenated?
* Relatedly, the goal of the work is to show that the attention-enabled agents come up with more compositional languages. For now a possible interpretation of Table1 is that such agents are simply more effective at solving the task (have higher acc) and, consequently, they attain better compositionality. Is there a way to disentangle success rate vs compositionality, ie showing that for a fixed success rate the languages would be more compositional? Maybe the baseline architecture is simply too weak due to the averaging?
* Stepping back, I think it is not surprising that a hand-crafted architecture, tailored for this specific game setup, can be created to promote compositionality in the emergent languages. What makes this particular architecture interesting? One specific way this architecture can be interesting is if it is very general and as powerful as standard Transformer yet provides additional benefit of compositionality. However, I believe the current experiments do not allow this conclusion, primarily because they only use the single-head attention. Would the compositionality bias persist with multiple heads?
* In Figure 3, we only have examples where Speaker and Listener have access to the same example. It would be interesting to look at the attention map when the examples are different but from the same label (is the position important?).
* For the Fashion-MNIST game: What is the size of the training dataset (it is only specified that there are 45 labels)?


* Typo: S2.4, "Positioanl"



**Summary Of The Paper:**

The paper studies languages that emerge when two agents (Speaker and Listener) communicate to solve a referential game. The authors propose to endow the agents with attention mechanisms to create a pressure for learning more compositional emergent languages. The experimental study shows that indeed changing the architectures in this way results into increases in the compositionality metrics (topographic similarity, pos-dis, and bos-dis). The authors also provide interesting insights in the behaviour of the agents by analysing the emerging attention patterns.

**Summary Of The Review:**

My main comments are:
* It is not clear that the baselines are not overly simplistic;
*  I think it is not surprising that a hand-crafted architecture, tailored for this specific game setup, can be created to promote compositionality in the emergent language. And it seems to me that this particular architecture is hand-crafted as it has a single-head attention. What makes this particular architecture interesting?
* Ideally, to support the claim of the paper, one might want to disentangle the task success rate from how compositional language is;

---

> ### Author Response · Authors · 2022-11-18
> **First Response to Reviewer 4G68**
>
> > The main novelty of this paper is agents’ architecture (section 2.2). However, this attention mechanism is not sufficiently explained. The authors state that the query is the symbol of the message and the keys/values are the inputs’ representation. This is still a vague description of the mechanism.
>
> Thank you for pointing out this.
> We have tried to improve the description of the model architecture in Section 2 of the new version.
> Appendix C includes the details we omit due to the space limit, we refer the readers.
>
> > The baseline architecture simply averages input embeddings. Is this the most fair baseline we can come up with? For instance, can we have an architecture where the set of vectors are projected to a smaller dimension (/2) and concatenated?
>
> Indeed, the suggested model here is a reasonable and potentially stronger baseline because we can assure that attributes of the input object are disentangled dimension-wise.
>
> We have added the comparison with the average-based and concatenation-based object encoders in Appendix D.
> Indeed, the concatenation-based model provides higher compositionality score.
> Yet, the attention models outperform the concatenation-based baseline by a large margin, so this does not affect the overall discussion.
>
> > Relatedly, the goal of the work is to show that the attention-enabled agents come up with more compositional languages. For now a possible interpretation of Table1 is that such agents are simply more effective at solving the task (have higher acc) and, consequently, they attain better compositionality. Is there a way to disentangle success rate vs compositionality, ie showing that for a fixed success rate the languages would be more compositional? Maybe the baseline architecture is simply too weak due to the averaging?
>
> Thank you for your suggestion.
> To clarify this point, we present the training accuracy on the training set in Figure 2 of the new version.
> In the one-hot game, most agents converge to >99% of success rate on the training set and this suggests the baseline agents are expressible enough for the task.
>
> For the Fashion-MNIST game, the attention models show better training accuracies.
> However, if we compare the Transformer NoAT-NoAT agents and the AT-AT agents, some of them exhibit similar training accuracy (around 75%) but the AT-AT agents provide significantly better TopSim scores (the minimum is around 0.45 while the maximum of the AT-AT agents is around 0.33).
> This indicates that the model flexibility indicated by the training accuracy is not a direct source of the improved compositionality of the attention agents.
>
>
> > Stepping back, I think it is not surprising that a hand-crafted architecture, tailored for this specific game setup, can be created to promote compositionality in the emergent languages. What makes this particular architecture interesting?
>
> The focus of our work is not to develop a model that improves compositionality, but rather proving that the inductive bias from the attention mechanism improves compositionality.
> The motivation for choosing the Transformer architecture was mainly its ubiquity in the ML field.
> To clarify this point and provide robust evidence, we have added the experimental results from the LSTM-based models, comparing non-attention and attention variants.
>
> > Would the compositionality bias persist with multiple heads?
>
> The reason for adopting the single-head attention is for ease of analysis rather than improving compositionality.
> Still, it would be interesting to see the effect of having the multi-head attention.
> We provide the results of the comparison of different number of attention heads in the one-hot game in Appendix E.
> As a result, we do not observe the advantage of the single-head attention in task performance and compositionality in this setting.
>
>
> > In Figure 3, we only have examples where Speaker and Listener have access to the same example. It would be interesting to look at the attention map when the examples are different but from the same label (is the position important?).
>
> Thank you for your suggestion.
> Indeed, it is important to showcase examples with different Speaker and Listener inputs to clarify what they learn.
> We have updated Figure 4 in the new version, showing the agents can communicate the item identities regardless of the positions.
>
>
> > For the Fashion-MNIST game: What is the size of the training dataset (it is only specified that there are 45 labels)?
>
> We have added the details in Appendix A.
> We used 5,000 images for each item class in the dataset, and then create image objects for the referential game by sampling from the image pool.
> We precompute 3,000 images and their features for each object class (\ie, each combination of two items), so we used 3,000 × 45 different images in total for training and evaluation.
>
> > Typo: S2.4, "Positioanl"
>
> Thank you. We have fixed the typo accordingly.

---

### Official Review · Reviewer_4TD5 · 2022-11-02

**Confidence:** 2
**Correctness:** 4
**Technical Novelty And Significance:** 2
**Empirical Novelty And Significance:** 2
**Recommendation:** 5

**Clarity, Quality, Novelty And Reproducibility:**

The paper was clear and reproducible. I do have some concerns about the novelty of the claims, which are explained in the *Weaknesses* section.

**Strength And Weaknesses:**

*Strengths*

The experimental results in this paper support the hypothesis that attention helps in the particular architecture considered. The improvements in all metrics over the ablated non-attention model are significant, despite large error bars.

*Weaknesses*

My primary concern is that this paper has two main contributions: 1) weaker models result in worse performance and 2) attention is interpretable. These contributions are relatively limited, as a correlation between model flexibility and performance is common and the interpretability of attention has been analyzed previously [1].

I am hesitant to ask for more experiments, but will suggest one. To increase the scope of the first contribution, one could test if more powerful models actually do result in task metric improvements. For example, one could run a multi-headed attention speaker and/or listener. It's possible that a single head attention model might perform better, since a compositional encoding may want to focus on one aspect of the referent at a time. This would contradict the "weaker models result in worse performance", and be an interesting additional contribution.

*Questions*

* The error bars in Table 1 are much larger than the ones reported in [2], which are run with a larger message space. I assume this is due to variance in the REINFORCE gradient estimator, which is one of the reasons for introducing the KL regularization. Is there an alternative explanation for this, and/or a way to reduce the size of the error bars? Perhaps other forms of variance reduction, such as multi-sample estimators and/or control variates.

[1] Wiegreffe, Sarah and Yuval Pinter. “Attention is not not Explanation.” EMNLP (2019).
[2] Chaabouni, Rahma, et al. "Emergent communication at scale." International Conference on Learning Representations. 2021.

**Summary Of The Paper:**

This paper analyze the effect of attention on emergent communication. The paper hypothesizes that the attention mechanism will bias agents towards more compositional encodings. Experimental results on a small-scale symbolic game and image reference game derived from Fashion-MNIST demonstrate that attentive speaker and/or listeners result in improvements in task success on unseen objects and various proxy measures of compositionality such as topographic similarity and positional disentanglement. Further analysis builds on the intepretability of attention, and shows that attention can be used to analyze attribute and message alignment, as well as predict coordination between speakers and listeners.

**Summary Of The Review:**

I recommend a borderline reject, since I am not confident that the contributions are novel.

---

> ### Author Response · Authors · 2022-11-18
> **First Response to Reviewer 4TD5**
>
> > 1) weaker models result in worse performance
> > These contributions are relatively limited, as a correlation between model flexibility and performance is common
>
> Thank you for your comment.
> One claim we are trying to make is that the attention mechanism improves the compositionality of emergent language.
> Indeed, the results provided in the previous version did not make it clear whether the improvement of compositionality comes from the flexibility of the attention models or not, i.e., the attention models may be just better at fitting to the data with more flexibility and that leads to better compositionality.
>
> To show that the source of the improved compositionality is not only the flexibility of the models, we present the training accuracy on the training set in Figure 2 of the new version.
> If the attention models are strictly more flexible than the non-attention models, they should exhibit better training accuracy.
> However, this is not necessarily the case.
> In Figure 2(a), the Transformer AT-AT model seems to struggle with fit to the training data compared to the NoAT-NoAT baseline but the AT-AT model still shows better TopSim scores (this is more obvious in the agents produce a longer sequence in Figure 7).
> This suggests that the attention models are not necessarily flexible in solving the task, perhaps even likely to fall into a local minimum during optimization by failing incorrect attention.
> Yet, the attention models exhibit higher compositionality scores, which supports the claim that one of the sources of the improved compositionality is the inductive bias provided by attention, rather than the freedom added by it.
>
> > To increase the scope of the first contribution, one could test if more powerful models actually do result in task metric improvements. For example, one could run a multi-headed attention speaker and/or listener. It's possible that a single head attention model might perform better, since a compositional encoding may want to focus on one aspect of the referent at a time. This would contradict the "weaker models result in worse performance", and be an interesting additional contribution.
>
> Thank you for the suggestion.
> This adds evidence to distinguish the point discussed above.
> As shown in Figure 10 of Appendix E, we find that increasing the number of attention heads does not significantly improve the model performance in the one-hot game.
> Due to the time constraint, we have only experimented with the one-hot game but there is still a possibility that the multi-head attention benefits in more complex environments.
> Yet, from the observations and discussion above, we think that the model flexibility only does not explain the improvements in compositionality.
>
>
> > 2) attention is interpretable
> > These contributions are relatively limited, ... the interpretability of attention has been analyzed previously [1].
>
> Our contribution lies in the analysis of the attention patterns that emerge within the referential game.
> How the patterns could emerge in the setting of the referential game has not been entirely clear.
> We hope that actually showing what patterns the emergent language exhibits serve as a step forward toward better understanding emergent language.
>
>
> > The error bars in Table 1 are much larger than the ones reported in [2], which are run with a larger message space. I assume this is due to variance in the REINFORCE gradient estimator, which is one of the reasons for introducing the KL regularization. Is there an alternative explanation for this, and/or a way to reduce the size of the error bars? Perhaps other forms of variance reduction, such as multi-sample estimators and/or control variates.
>
> We think that the variance can be decomposed into two parts (1) optimization instability (2) intrinsic ambiguity of the solution space.
> As you suggested, we agree that optimization instability can be mitigated by incorporating advanced techniques for optimization.
> Besides that, we think a large part of the variance also comes from the intrinsic ambiguity of the solution space.
> The agents are trained with only a part of possible object combinations, which does not assure how well they perform for unseen object combinations.
> The variance observed in our setting is larger than the ones in [2] because, we think, our setting uses strictly unseen combinations of attributes for evaluation whereas [2] uses randomly split natural images.
> The variance of the generalization accuracy and TopSim score with the simple one-hot game can be observed in other literature such as [3].
>
>
> [1] Wiegreffe, Sarah and Yuval Pinter. “Attention is not not Explanation.” EMNLP (2019).
>
> [2] Chaabouni, Rahma, et al. "Emergent communication at scale." International Conference on Learning Representations. 2021.
>
> [3] Chaabouni et al, "Compositionality and Generalization In Emergent Languages" EMNLP (2020)

---

### Author Response · Authors · 2022-11-18
**General Response to Reviewers**

We would like to thank all the reviewers for their time and constructive feedback.
We have updated the paper as follows:

- added experimental results of LSTM-based agents.
- added experimental results of the one-hot game with different configurations in Appendix.
- tried to add more clarity to the architecture and task description.
- removed the discussion about the PosDis and BosDis metrics.
- added some experiments suggested in the reviews in Appendix (see individual replies).

Please find answers to individual comments and questions below.

---

### Decision · Program_Chairs · 2023-01-20

**Decision:**

Reject

**Justification For Why Not Higher Score:**

Insufficient experiments w.r.t analysis (on how attention really helps), baseline/model architectures, and the design of the evaluation tasks.

**Justification For Why Not Lower Score:**

-

**Metareview: Summary, Strengths And Weaknesses:**

This paper studies languages that emerge when two agents (Speaker and Listener) communicate to solve a referential game and explores how attention may impact language emergence, especially towards compositionality. Experimental results on a small-scale symbolic game and image reference game demonstrate that attentive speakers and/or listeners result in improvements in task success on unseen objects and various proxy measures of compositionality.

The paper is well-written and the experimental results support the hypothesis that attention helps in the particular architecture considered.

However, the primary concerns from the reviewers are on the experimental evaluation and analysis, including 1) how to prove the attention really helps compositionality, rather than just being a more flexible model architecture (Reviewer 4TD5, Reviewer kyNy, Reviewer 4G68). 2) using a hand-crafted architecture/baseline and lack of extensive architecture evaluated (e.g., depth, encoder/decoder, decoder-only) (Reviewer gZp6, Reviewer 4G68). 3) The toyish design of the evaluation tasks (Reviewer gZp6, Reviewer kyNy).

Although the authors addressed some of the reviewers’ questions, the major concerns are not sufficiently addressed. In conclusion, we suggest a rejection.